# Effect of Shear Flow on Nanoparticles Migration near Liquid Interfaces

**DOI:** 10.3390/e23091143

**Published:** 2021-08-31

**Authors:** Ali Daher, Amine Ammar, Abbas Hijazi, Lazhar Benyahia

**Affiliations:** 1LAMPA, ENSAM Angers, 2 Boulevard du Ronceray, BP 93525, CEDEX 01, 49035 Angers, France; amine.ammar@ensam.eu; 2Faculty of Sciences1, MPLAB, Lebanese University, Beirut 6573, Lebanon; abhijaz@ul.edu.lb; 3Institut des Molécules et Matériaux du Mans (IMMM), UMR 6283 CNRS—Le Mans Université, Avenue Olivier Messiaen, CEDEX 09, 72085 Le Mans, France; lazhar.benyahia@univ-lemans.fr

**Keywords:** liquid–liquid interface, shear rate, nanoparticles, diffuse interface, phase field method, molecular dynamics, numerical simulation

## Abstract

The effect of shear flow on spherical nanoparticles (NPs) migration near a liquid–liquid interface is studied by numerical simulation. We have implemented a compact model through which we use the diffuse interface method for modeling the two fluids and the molecular dynamics method for the simulation of the motion of NPs. Two different cases regarding the state of the two fluids when introducing the NPs are investigated. First, we introduce the NPs randomly into the medium of the two immiscible liquids that are already separated, and the interface is formed between them. For this case, it is shown that before applying any shear flow, 30% of NPs are driven to the interface under the effect of the drag force resulting from the composition gradient between the two fluids at the interface. However, this percentage is increased to reach 66% under the effect of shear defined by a Péclet number *Pe* = 0.316. In this study, different shear rates are investigated in addition to different shearing times, and we show that both factors have a crucial effect regarding the migration of the NPs toward the interfacial region. In particular, a small shear rate applied for a long time will have approximately the same effect as a greater shear rate applied for a shorter time. In the second studied case, we introduce the NPs into the mixture of two fluids that are already mixed and before phase separation so that the NPs are introduced into the homogenous medium of the two fluids. For this case, we show that in the absence of shear, almost all NPs migrate to the interface during phase separation, whereas shearing has a negative result, mainly because it affects the phase separation.

## 1. Introduction

In recent years, using nanoparticles (NPs) in industrial and medical markets has grown significantly. They have been of immense significance in different branches of science and engineering. This is basically due to their unique properties, such as augmented reactivity and special optical properties, which make them very suitable for products and applications in tissue engineering, composite technology, enhanced oil recovery and drug delivery [1,2]. In addition, NPs arise in nanoparticle-armored fluid droplets [3] and phase-arrested gels [4]. It is well known that studying biological processes on the nanoscale level is an essential point behind the development of nanotechnology [5].

The assembly of NPs at the liquid–liquid interface is essential in the preparation and stabilization of conventional emulsions, which have wide applications in petroleum, cosmetics, food, and biological transferring [6,7]. Modeling the dynamics of NPs at liquid–liquid interfaces has a crucial role in developing static and dynamic flow models that help in drug delivery and understanding the biological and physical phenomena inside the cells of the body [8].

In addition, the dynamics of NPs in non-aqueous media, such as ionic liquids (ILs), was reported in many studies. It was shown that the design and preparation of the nanomaterials are well planned and executed, using ILs to produce tunable and functional fluid ILs-based nanomaterials, and it also was reported that ILs help to synthesize nanomaterials with various functionalized surfaces [9].

Furthermore, the non-extensivity of entropy was investigated for different sizes of colloidal Ag nanoparticles (NPs), and it was shown that the subextensivity of entropy occurs for colloidal Ag NPs. In the small size of colloidal Ag NPs and at low temperature, nonextensivity is important [10]. Taherkhani et al. used classical molecular dynamics (MD) simulations to investigate the radial distribution, glass transition, ionic transfer number, and electrical conductivity of the ionic liquid 1-ethyl-3-methylimidazolium hexafluorophosphate [EMIM][PF6] ionic liquid encapsulated in carbon nanotube (CNT), and they also studied the effect of nitrogen as a doping element in CNT on these properties of [EMIM][PF6] by MD simulation. It was shown that in the presence of nitrogen, ion transfer uses a hydrogen bonding mechanism, while in its absence, ion transfer uses a diffusion mechanism in which the cation has a significant effect on the ion transfer and electrical conductivity [11].

The behavior of NPs is strongly affected by the surrounding environmental factors and thus, external effects will modify their dynamic properties. Many researchers have studied the effects of external fields on the aggregation of NPs. The effect of shear on nanoparticle dispersion in polymer melts was investigated by Karla et al. [12], and it was shown that shear significantly slows down the aggregation of NPs and such an effect is strongly dependent on the polymer chain length and shear rate. In addition, Karla et al. [13] studied how the NPs disperse in a block copolymer system under shear flow; they found that shear can have a pronounced effect on the location of NPs in block copolymers and that it can be used as a parameter to control nano-composite self-assembly. In addition, Minh D Vo et al. [14] used dissipative particle dynamics (DPD) methods to study the effect of shear and particle shape on the physical adsorption of a polymer on carbon nanoparticles; they found that there are three possible states of the polymer adsorption on carbon nanoparticles (adsorbed, shear affected, and separated states) depending on the value of the shear rate. Besides that, the effects of shear stress on the intracellular uptake of NPs in a biomimetic microfluidic system were investigated by Kang et al. [15], and they showed that for the case of cationic NPs, as the magnitude of the shear stress increases, the intracellular uptake of such NPs maximizes at a certain value of shear stress and then decreases gradually, which ensures that the shear stress has a crucial role in various nanoparticles and drug delivery systems.

Plater et al. investigated experimentally the effect of viscosity ration of polypropylene/Poly-e caprolactone blends on the localization of carbon black aggregates. The authors reported that the particles were dragged to the viscous phase, even when the particles were located initially into the more fluid phase, although they preferred to locate in the latter [16].

Becu et al. succeeded in visualizing a single armored droplet with nanoparticles undergoing a shear flow in another Newtonian medium. The results showed a continuous but clear slowdown of the droplet relaxation after successive strain jumps. This effect is related to the densification of the droplet interface by NPs when deformed [17].

In the current study, we focus on how shear force affects the migration of NPs to the liquid–liquid interface, which will help to understand how NPs behave under standard industrial processing conditions. The goal of this study is to implement a compact model for the simulation of the shear effect on the migration of spherical NPs near a liquid–liquid interface. Our work is based on the phase field method (PFM) for the fluids modeling, using the diffuse interface model, and molecular dynamics (MD) for modeling the motion of the nanoparticles, through which we superimpose the discrete model of NPs (using MD) on the continuum model of fluids (using PFM), which is a new idea in numerical modeling that we discussed briefly in our previous paper [18].

The content of this paper is as follows. In Section 2, we give details of the models and methods that we used in numerical simulation. We describe the diffuse interface model and give a brief description of the numerical implementation and time discretization. In Section 3, we discuss the numerical results for the migration of nanoparticles at the liquid–liquid interface. We investigate the effect of shearing on the migration of nanoparticles at the liquid–liquid interface, and we compare the results for different shear rates and different shearing times. Finally, a conclusion follows in Section 4.

## 2. Materials and Methods

### 2.1. Particle–Particle and Fluid–Particle Interactions

In this section, the discrete dynamic of particles is described. Molecular dynamics is the method of choice when one wants to study the dynamical properties of a system in full atomic detail, provided that the properties are observable within the time scale accessible to simulations. Time scale is one of the two main limitations of the method as will be discussed later. Molecular dynamics simulations are also useful when the system cannot be studied by the experimental methods mentioned above, for example, when the protein cannot be crystallized or is too big or insoluble to be studied by NMR.

To calculate the dynamics of the system, that is, the position of each atom as a function of time, Newton’s classical equation of motion is solved iteratively for each atom.

Each NP is considered a rigid spherical body whose velocity $\left(v_{i}\right)$ and position $\left(x_{i}\right)$ are updated by using Newton’s equation of motion, which relates the applied forces with the particle’s acceleration $\left(a_{i}\right)$ according to the following equation:(1)$$F_{i}\left(t\right)=m_{i}\;\frac{d^{2}x_{i}}{dt^{2}}=m_{i}a_{i}\left(t\right)$$
where $F_{i}$ is the applied force on the particle ‘i’, and $m_{i}$ is the mass of the particle. The applied forces can be classified into particle–particle interaction forces and external forces due to the fluid (in our case, we consider drag forces, Brownian forces and shear effects).

The force on each atom is the negative of the derivative of the potential energy with respect to the position of the atom: (2)$$f_{j-i}=-\nabla V\left(r\right)$$

If the potential energy of the system is known, then, given the coordinates of a starting structure and a set of velocities, the force acting on each atom can be calculated and a new set of coordinates generated, from which new forces can be calculated. Repetition of the procedure will generate a trajectory corresponding to the evolution of the system in time. The accuracy of the simulations is directly related to the potential energy function used to describe the interactions between particles. In molecular dynamics, a classical potential energy function is used that is defined as a function of the coordinates of each of the atoms. In macroscopic systems, the fraction of the particles near the wall is negligible, whereas in the MD simulations, this fraction is more significant, and the surface effect is of great importance. In order to reduce the surface effect and conserve the number of particles in the simulation box, periodic boundary conditions must be used so that when a particle enters or leaves the simulation region, an image particle leaves or enters this region.

The position and velocity of the NPs (*xi and vi* respectively) at each new time are calculated, using the velocity Verlet algorithm time integration scheme: (3)$$x\left(t+\delta \right)=x\left(t\right)+v\left(t\right)\delta +\frac{1}{2}\;a\left(t\right)\delta^{2}$$
(4)$$v\left(t+\delta \right)=v\left(t\right)+\frac{1}{2}\; [a\left(t\right)+a\left(t+\delta \right)]\delta^{2}$$
where δ indicates the time increments for the molecular dynamics (MD).

In our model, the particle–particle interaction is calculated, using the truncated Lennard–Jones ($LJ_{trunc}$) potential; this potential is used in many numerical studies in order to model particle–particle interactions. See [19] for example.
(5)$$V_{ij{}_{trunc}}^{LJ}\left(r\right)=\left\{\begin{array}{l} V_{LJ}\left(r_{ij}\right)-V_{LJ}\left(r_{c}\right)\;\;\;\;\;\;for\;r<r_{c}\\ 0\;\;\;\;\;\;\;\;\;\;\;\;\;\;\;\;\;\;\;\;\;\;\;\;\;\;\;\;\;\;\;\;\;\;for\;r>r_{c} \end{array}\right.$$

With
(6)$$V_{LJ}\left(r\right)=4\epsilon \; [{\left(\frac{\sigma }{r}\right)}^{12}-{\left(\frac{\sigma }{r}\right)}^{6}]$$
where $r_{c}$ is the cut-off distance, which is taken to be 3σ, ϵ is the depth of the potential well, σ denotes the equilibrium distance, and r is center to center separation between two particles.

Thus, the interaction force acting on the i−th particle induced by the j−th particle is given by the following:(7)$$f_{j\to i}=-\nabla V_{LJ_{trunc}\;}\left(r\right)\;n_{j\to i}=-\frac{\partial V_{LJ_{trunc}}\left(r_{ij}\right)}{\partial r_{ij}}n_{j\to i}$$
where $r_{ij\;}=r_{j}-r_{i\;}$ is the separation distance between two nanoparticles, i and j correspond to different NPs indices, and $n_{j\to i}$ represents the unit vector that point from $x_{j}$ to $x_{i}$.

So, for N particles, the force acting on each particle is formed by the individual interactions with all the neighboring particles:(8)$$F_{i}=\overset{N}{\underset{j\neq i}{\sum }}f_{j\to i}$$

The external forces acting on the NPs are due the surrounding fluids and thus, each NP is affected by the Brownian force $F_{r}$ and the drag force $F_{d}$ given by the following formulas:(9)$$F_{r}\left(t\right)=\sqrt{2\;D\;\Delta t}\;\chi$$
where Δt is the time step in the numerical model, D is the diffusion coefficient and χ is a normal random number whose average is zero and variance is one.

The drag effects are considered by the following:(10)$$F_{d}=\varphi \;\left(v_{f}-v_{p}\right)$$
where φ is the drag coefficient, $v_{p}$ is the particle’s velocity and $v_{f}$ is the fluid’s velocity resulting from the Navier–Stokes equation involving the concentration gradient (see [20] for details):(11)$$v_{f}=div\left(\nabla c\;⊗\;\nabla c\;\right)$$
where ⊗ corresponds to the tensor product. The definition of c is introduced in the next section. It denotes the phase field scalar used to describe the mixture.

### 2.2. Diffuse Interface Model

In the diffuse interface model, the convective Cahn–Hilliard equation is given by the following:(12)$$\frac{\partial c}{\partial t}+u.\nabla c-\nabla .j_{A}=0$$

The diffusive flux is given by $j_{A}=M\nabla \mu$, where M denotes the mobility (scalar in the case of isotropic separation mixture, and tensor in the case of non-isotropic separation). μ is the chemical potential obtained from the variational derivative of the free energy with respect to the mass fraction c.

The Cahn–Hilliard theory [21] assumes that the driving force for diffusion is the gradient of the chemical potential, and thus, the above equation is generally written as follows:(13)$$\frac{\partial c}{\partial t}+u.\nabla c=\nabla .\left(M\nabla \mu \right)$$

The free energy is a double-sink function that implies that the only stable equilibrium values of *c* are +1 or −1. We use the classical symmetric form of ψ, which is used in most cases. However, other forms of non-symmetric potential can be used. The expression of free energy is assumed given by the following:(14)$$f\left(c,\nabla c\right)=-\frac{1}{2}\;\alpha c^{2}+\frac{1}{4}\;\beta c^{4}+\frac{1}{2}\;\varepsilon {\left|\nabla c\right|}^{2}$$
where ε is the gradient energy parameter and α and β are positive constants. Thus, we have the following: (15)$$\mu =\frac{\delta f}{\delta c}=-\alpha c+\beta c^{3}-\varepsilon \nabla^{2}c$$

For this study, we apply shear forces. In order for the expression of this force to be compatible with our framework of periodic study, we consider that the shear force is taken to be periodic along the y-axis and defined by the following:(16)$$f_{shear}=\frac{\sin2*\pi *y}{l}$$

This is illustrated in Figure 1.

The effect of shearing is introduced into the momentum equation represented by the Navier–Stokes equation (NS) with a phase field-dependent surface force as follows [22]:(17)$$\rho \;\left(\frac{\partial u}{\partial t}+\left(u.\nabla \right)u\right)=-\rho \nabla g+\nabla .\eta \;\left(\nabla u+\nabla u^{T}\right)+\rho \;\mu \nabla c+f_{shear}$$

We also consider the continuity equation for an incompressible fluid as follows:(18)∇.u=0

In the above equations, the dynamic viscosity of the fluid is denoted by η, u is the velocity field and g is the Gibbs free energy given by the following:

g=f+p/ρ, where p is the pressure and ρ is the mass density. The superscript T stands for the transpose operator.

### 2.3. Scaling the Equations

In order to simplify the equations and minimize the effects of round-off errors, it is preferable to use a set of dimensionless parameters. So, we scale the governing equations by defining $U_{c}$ and $L_{c}$ as the characteristic velocity and characteristic length, respectively, $T_{c}=\frac{L_{c}}{U_{c}}=\sigma .\;\sqrt{\frac{m}{\epsilon }}$ as the characteristic time and $\varepsilon_{c}$ as the characteristic energy. The characteristic length of the phase field scale is related to that of the molecular dynamics as $L_{c}=100\;\sigma$ (fixed ratio chosen for our studies).

We introduce $r^{*}=r/\sigma$, $\varepsilon^{*}=\varepsilon /\varepsilon_{c}\;$, $c^{*}=\frac{c}{c_{B}}$, $u^{*}=\frac{u}{U_{c}}$, $t^{*}=\frac{tU_{c}}{L_{c}}$, $\mu^{*}=\frac{\mu \xi^{2}}{\varepsilon c_{B}}$, $\eta^{*}=\frac{\eta }{\eta_{c}}$ as the normalized viscosity.

In addition, the dimensionless drag coefficient is defined as follows:$$\varphi^{*}=\frac{\varphi \;\sigma }{\sqrt{\varepsilon_{c}\;m}}$$

The Péclet number is defined as the product of a shear rate by a characteristic time as follows: $$\mathrm{Pe}=\dot{\gamma }.\;\;T_{c}=\dot{\gamma }.\;\sigma .\;\sqrt{\frac{m}{\epsilon }}$$

$\xi =\sqrt{\frac{\varepsilon }{\alpha }}$ Is the interfacial thickness and $c_{B\;}=\sqrt{\frac{\alpha }{\beta }}$ is the bulk concentration, which represents the mean field equilibrium value. Dropping the asterisk notations, we obtain the following:(19)$$\frac{dc}{dt}=-u.\nabla c+\frac{1}{P_{e}}\;\nabla^{2}\mu$$
(20)$$\mu =c^{3}-c-C\nabla^{2}c$$
(21)∇.u=0
(22)$$\nabla g-\nabla .\eta \;\left(\nabla u+\nabla u^{T}\right)=\frac{1}{C\;.\;C_{a}}\;\mu \nabla c+f_{shear}$$
(23)$$V_{LJ}\left(r\right)=4\epsilon \; [{\left(\frac{1}{r}\right)}^{12}-{\left(\frac{1}{r}\right)}^{6}]$$

From this potential, we can get the force to be the following:(24)$$F_{LJ}=-\nabla V_{LJ\;}\left(r\right)$$
(25)$$F_{d}=\varphi *\;\left(v_{f}-v_{p}\right)$$

We obtain the following set of dimensionless groups: the Cahn number C and the Capillary number $C_{a}$, defined respectively as follows:$$C=\frac{\xi }{L_{c}}\;;\;C_{a}=\frac{\xi \eta U_{c}}{\rho \varepsilon c_{B}{}^{2}};$$

### 2.4. Numerical Implementation

To model the dynamics of the NPs, we use the molecular dynamics (MD) method with periodic boundary conditions implemented in order to conserve the number of NPs in the simulation box.

In addition, we use the phase field method in order to model the two fluids and the formation of the interface between them. In this method, the concentration is defined as in Equation (19).

To find the weak form, we multiply by a weighting function $c^{*}$ and integrate over the whole fluid domain Ω to obtain the following: (26)$$\int c^{*}\;\frac{dc}{dt}\;dΩ+\int c^{*}\;u.\left(\nabla c\right)dΩ-\int \frac{1}{P_{e}}c^{*}\;\nabla^{2}\mu \;dΩ=0$$
where $c^{*}={\underline{c}}^{*}{\;}^{T}\underline{N}$ and N is defined as $[\begin{array}{cccc} N_{1} & N_{2} & N_{3} & N_{4} \end{array}].$

In the above equation $N_{1},\;$… $N_{4}$ are the quadratic interpolation functions of the 4-node quadrilateral element:(27)$$\frac{dc}{dt}={\underline{N}}^{T}\;\dot{\underline{c}}$$

The finite element interpolation for the gradient of the concentration is described in terms of the linear combination of the shape function derivatives, given in matrix form, by the following: (28)$$\nabla c=B\;\underline{c}$$
$$B= [\begin{array}{cccc} N_{1,x} & N_{2,x} & N_{3,x} & N_{4,x}\\ N_{1,y} & N_{2,y} & N_{3,y} & N_{4,y} \end{array}]$$

Solving Equation (23), we get:(29)$${\underline{c}}^{*}{\;}^{T}\int \underline{N}\;{\underline{N}}^{T}\;dΩ\;\underline{\dot{c}}+{\underline{c}}^{*}{\;}^{T}\;\int \underline{N}\;\underline{u}.\;B\;dΩ\;\underline{c}-{\underline{c}}^{*}{\;}^{T}\frac{1}{P_{e}}\;\int N\;\nabla^{2}\mu \;dΩ=0$$

This integration allows obtaining a linear system that has to be solved at each time step, which can be solved using a semi-implicit or explicit time integration scheme:(30)$$M\;\dot{\underline{c}}+G\;\underline{c}+F\left(\underline{c}\right)=0$$

Similarly, in order to solve the velocity equation, we can write the following:(31)$$\int u^{*}\nabla^{2}u\;dΩ+\int u^{*}A\mu \nabla cdΩ=0$$
(32)$$\mathrm{where}\;u^{*}={\underline{u}}^{*}{\;}^{T}\underline{N}$$
(33)$$\mathrm{and}\;\nabla^{2}u=K\;\underline{u}$$

Solving Equation (28), we obtain the following:(34)$${\underline{u}}^{*}{\;}^{T}\int N\;K\;dΩ\;\underline{u}+{\underline{u}}^{*}{\;}^{T}\;\int N\;A\;\mu \;B\;dΩ\;\underline{c}\;=0$$

This integration gives the following linear system to be solved at each time step:(35)$$T\;\underline{u}+H\;\underline{c}=0$$

The position and velocity of the NPs ($x_{i}\;and\;v_{i}$ respectively) at each new time are updated, using the velocity Verlet integration scheme.

## 3. Results and Discussion

### 3.1. Fluids Separation and Interface Formation

The system is composed of two liquids that are normally immiscible. Due to some external effects (temperature for example), these two liquids may be mixed to form one thermodynamic phase. So, we can start the study from an initial state where the two fluids are totally mixed as shown in Figure 2 (left figure). 

These figures represent a 2D plot of the simulation box representing the two fluids.

The side bar in Figure 2 represents the evolution of the concentration between the two fluids in order to form the interface. The concentration is taken to vary from −1 in the first fluid (blue) to 1 in the second one (red). In our model, the area fractions of the two phases are taken, as shown in Table 1.

As time goes on, and since the two liquids are normally immiscible, the molecules of each fluid immediately start to cluster together into microscopic clusters throughout the liquid. These clusters then rapidly grow and coalesce until we obtain an equilibrium state in which the two fluids are totally separated, and the interface is formed between them, as shown in Figure 2 (right).

The blue medium represents the first liquid, and the red represents the second liquid;the concentration is varying, according to the diffuse interface model. Note that our simulation box is bounded between 0 and 1 along the two axes as shown by the limiting lines in the figure, but we represent a periodic repetition of this box in Figure 2 (right) for clarity; this is done in all 2D figures throughout the paper.

### 3.2. Introduction of Nanoparticles after the Separation of the Two Fluids and the Formation of the Interface

After the two fluids are separated and the interface is formed between them, we introduce N nanoparticles randomly into the system.

We consider two cases regarding the concentration of NPs in the medium (total area of the NPs relative to the area of the medium of two fluids). The first case is 0.06 (200 NPs), and the second is 0.3 (1000 NPs).

#### 3.2.1. Low Concentration of Nanoparticles

Consider 200 NPs (concentration of NPs 0.06) distributed randomly within the two fluids as shown in Figure 3.

Neglect particle–particle interactions

For the first moment, let us neglect the interaction between the NPs via LJ potential and study the migration of NPs toward the interface. In this part, we examine the behavior of the NPs once they are introduced into the mixture of the two fluids that is already separated and the interface is formed between them. The main goal here is to study whether the external shear effect can improve the migration of NPs to the interface or not. Different cases regarding the shear effect are simulated. We start with the case involving no shear, and then we increase the Péclet number progressively; in each case, the percentage of NPs migrating to the interface is plotted.

No shear case.

Once introduced into the medium of the two fluids, the NPs near the interface are affected by the drag force given in Equation (7), resulting from the concentration gradient between the two fluids at the interface. These NPs are driven to the interfacial region, whereas the NPs far from the interface are not affected by this force, and thus, only 37% of the NPs migrate to the interface as shown in Figure 4.

In the simulation, time is defined as a dimensionless parameter (normalized time) $t^{*}=\frac{{\mathrm{tU}}_{c}}{L_{c}}$ as shown in Section 2.3 (scaling the equations).

The percentage of NPs belonging to the interface is calculated by finding the number of NPs belonging to the interfacial region relative to the total number of NPs introduced into the medium.

In this work, the interface is defined as the region of high concentration gradient, and we are able to track the position of the NPs and determine those that reach the interfacial thickness by calculating the concentration gradient at every position. By this way, we are able to determine whether each NP reaches the interface or not. It is important to note that the noise in Figure 4 (right) and all the coming figures are mainly due to the effect of the Brownian force introduced in the system. Although the effect of the other forces dominates the effect of the Brownian force, there are still some effects as seen by the percentage of error caused due to the Brownian force After the steady state is reached after introducing NPs, a shear is applied for a certain duration (T-shear), defined relative to the characteristic time. Thus, under the effect of the drag force, NPs near the interface are adsorbed. We consider different cases for which we quantify the evolution of the NPs percentage belonging to the interfacial region.

2.Simulation with shear rate = 0.4; *Pe* = 0.008, T-shear = 0.3.

In Figure 5, MD corresponds to molecular dynamics simulation, and it represents the separation of the two fluids in the absence of shear (before applying shear). This figure shows that the percentage of NPs migrating to the interface in the absence of shear is about 37%, and this percentage is not significantly affected in the case of low shear (low Péclet number *Pe* = 0.008). 

For this case, we find that a low Péclet number does not have a noticed effect on the percentage of NPs belonging to the interface.

3.Simulation with shear rate = 7.7; *Pe* = 0.154, T-shear = 0.3.

Increasing the shear rate and thus, increasing the Péclet number to 0.154, the percentage of NPs belonging to the interfacial region increases under the effect of shear, from 37% to about 60% as shown in Figure 6.

4.Simulation with shear rate =15.8; *Pe* = 0.316, T-shear = 0.3.

In Figure 7, it is clear that increasing the shear rate and thus, increasing the Péclet number, enhances the migration of the NPs toward the interface to reach about 62% at the end of the shear; also, this percentage increases a little bit to reach 70% after stopping shearing. 

This is mainly due to the fact that as the shear rate increases, more interfaces are formed in the medium (i.e., the length of the interface is increasing) and thus, the percentage of NPs belonging to the interfacial region increases. After stopping shearing, the interfaces tend to reach an equilibrium state, and they reach the separation phase. In this case, the NPs that are still near the interface are attracted to the interfacial region, due to the concentration gradient so that the percentage increases a little bit to reach 70%.

Including particle–particle interactions.

In this part, we consider the same situation discussed above (concentration of NPs 0.06), but this time we take into account the particle–particle interactions.

Once introduced, the NPs near the interface are driven by the hydrodynamic drag force to the interfacial region, and all the NPs that are close to each other interact by the particle–particle interaction force, so the NPs form clusters around each other as shown in Figure 8.

In this part, we also study the effect of different shear rates.

1.No shear case.

Evaluating the percentage of NPs that are within the interfacial thickness, before applying any shear on the system, shows that this percentage is about 26% as shown in Figure 9. This value is smaller than that found in the case of neglecting the Lennard–Jones potential. This is mainly because the formation of clusters prevents the clustering NPs from reaching the interface. In reality, they agglomerate around the ones that are in the interfacial thickness so that the cluster is attached to the interface by some of the NPs forming it.

The distribution of the NPs at the interface and the accumulation of the others over them are shown in Figure 8 above. In this case, the NPs that are attached to those at the interface are seen in the figure to belong to the first fluid (red) or the blue one (blue) and thus, they are not considered to belong to the interface. They are blocked by the ones that reached the interface before.

Now, we apply shear after the NPs near the interface are adsorbed, and we investigate the effect of shear on the migration of NPs to the interfacial region.

2.Simulation with *Pe* = 0.008; T-shear = 0.3.

As for the case without the Lennard–Jones potential, small shear rates do not have an important influence on the system. Thus, the percentage of NPs belonging to the interfacial region is not significantly modified, compared to the case of no shear as shown in Figure 10.

3.Simulation with shear rate =15.8; *Pe* = 0.316; T-shear = 0.3.

As the shear rate increases, more interfaces are formed in the medium (i.e., the length of the interface is increasing) and thus, the percentage of NPs belonging to the interfacial region increases, as shown in Figure 11.

One can notice that the percentage in these cases differs from that in the cases discussed above, where we neglected the Lennard–Jones potential. This is mainly due to the fact that in the case when the particle–particle interactions are taken into account, the NPs accumulate around each other and form clusters at the interface.

It is noticed in the literature that embedding particles at the liquid interfaces may, for example, lead to increased stability of biphasic systems, such as Pickering’s emulsions [23], or lead to a double percolation morphology followed by electrical conductivity as with immiscible polymer blends [24]. For the latter case, several studies have focused on the competition or synergy between thermodynamics and hydrodynamics that is inerrant to mixing processes [25,26,27,28] on the particle localization. However, the observation of the particle adsorption dynamics remains rarely considered. For example, Keal et al. were able to demonstrate, by confocal microscopy tracking, the adsorption by natural diffusion of colloidal particles at a liquid–liquid interface of very low interfacial tension. To date, we are not aware of any experimental work describing the adsorption dynamics of nanoparticles at the liquid–liquid interfaces under flow.

The blue boxes (marked area) in Figure 12 show that the two interfaces come so close, and the particles accumulate between them.

#### 3.2.2. Simulation with High Concentration of NPs

As for the previous concentration, we take into account two cases. First, we neglect the Lennard–Jones (L.J.) potential and then we include it in the second case.

Simulation of neglecting the L.J.potential.

For the initial state, consider 1000 NPs (concentration of NPs 0.3) randomly distributed within the two fluids, as shown in Figure 13a.

First, neglecting the effect of the L.J. potential, we discuss the effect of different shear rates applied for the same duration (T-shear), and in addition, we study the effect of different shearing durations for the same shear rates. i.e., for the same Péclet numbers.

By increasing the Péclet number to 0.316, we find that about 66% of the NPs are driven to the interfacial thickness for shear duration (T-shear = 0.3). Thus, we again ensure that shearing enhances the migration of the NPs toward the liquid–liquid interface.

Simulation including the L.J. potential.

In this part, we include the effect of the Lennard–Jones potential, for the case of 1000 NPs within the simulation box. We fix *Pe* = 0.3160 and T-shear = 0.3, and we examine different cases through which we vary certain parameters related to the L.J. potential.

1.ϵ=1, σ=0.02

In Figure 14, it is clear that by increasing the concentration of NPs within the simulation box, each NP has an interaction with all the NPs close to it. Due to the high concentration of NPs, all of them are connected with each other and thus, the L.J. force is strong enough to prevent the shear from affecting the motion and the location of the NPs.

2.Simulation with ϵ=0.1, σ=0.02

In experimental studies, it is possible to control and vary the value of the potential between each NP and its neighbors and thus, it is possible to modify the particle–particle interactions in order to match some industrial needs [29,30].

In order to study the effect of shearing in such cases, we minimize the value of the potential well’s depth, ϵ, in the definition of the L.J. potential.

The results are shown in Figure 15, where we find that in such cases, shearing affects the motion of the NPs and let them migrate toward the interfacial region. In addition, there are some NPs connected between two adjacent interfaces, and this matches the experimental results reported by Becu and Benyahia [14].

It is clear from the obtained results that by controlling the value of the potential between neighboring NPs and under the effect of shear, we are able to let the NPs form a layer on the interface, even when the concentration of the NPs in the fluids is high.

### 3.3. Introduce the NPs Randomly into the Mixture of the Two Fluids before Phase Separation

In this part, we study the case of NPs introduced into the medium of the homogenous mixture of the two fluids before phase separation and thus, before the formation of the interface as shown in Figure 16.

3.Simulation with no shear.

As time goes on, and since the two fluids are immiscible, they will start to separate, but this time the NPs are within the fluids and thus, their motion is affected by the phase separation.

In the absence of shear, it is clear from Figure 17 that almost all the NPs (100%) are driven to the interfacial region after a time of 2.2 relative to the characteristic time. The percentage increases progressively during the phase separation so that the NPs are affected by the force resulting from the concentration gradient throughout all the phase separation time. 

Comparing these results to the case when the NPs are introduced after phase separation has occurred, it is clear that the time of NPs introduction is important.

Since the percentage of NPs adsorbing at the interface is less when the phase separation is already occurred, this seems to indicate that the potential of adsorption is higher in this case. However, when we start from the homogenous state, the potential barrier is less, and thus, more particles can be adsorbed at the interface.

In order to quantify the effect of shear on the migration of the NPs to the interface in this case, we present the results obtained for different shear rates applied to the system just at the beginning of the simulation before the phase separation starts.

4.Simulation with shear rate = 0.4; *Pe* = 0.008

As for the previous cases, a small shear rate does not have an important effect on the obtained results, compared to the case of no shear as shown in Figure 18.

5.Simulation with shear rate 7.7; *Pe* = 0.154.6.Simulation with shear rate 15.8, *Pe* = 0.316

Increasing the shear rate will have a negative result with respect to the migration of NPs towards the interface. As seen in Figure 19 and Figure 20, the percentage of NPs belonging to the interface is smaller compared to the case of no shear for all time durations.

So as seen in the figures above, we find that in the case when we introduce the NPs into the medium of the two mixed fluids before phase separation takes place, more NPs will migrate to the interface in the absence of shear.

## 4. Conclusions

In this study, we have investigated the effect of shear force on the migration of nanoparticles toward the interface of two immiscible liquids.

We have implemented a numerical simulation through which we modeled the two fluids using the diffuse interface model. Two cases were investigated. The first one is to introduce the NPs randomly into the medium of two fluids that have been separated and the interface was formed between them. The second case is to introduce the NPs randomly into the mixture of the two fluids before phase separation takes place. For the first case, if we leave the fluids without any external effect, we find that only a small percentage, not greater than 30% of the NPs will migrate towards the interface. These NPs are the ones that are close to the interface when we introduce them randomly into the medium. Their migration toward the interface is mainly due to the effect of the drag force related to the concentration gradient in the interfacial region.

We have found that inducing a shear onto the system enhances the percentage of NPs belonging to the interface. It has been shown that there are two factors affecting the percentage of NPs migrating towards the interface, the value of the shear rate modified using the Péclet number, *Pe and* the shear duration T-shear. Introducing a shear defined by *Pe* = 0.154 for a duration of 0.63 drives 66% of the NPs to the interface compared to 30 % for the case of no shear effect. The same result was obtained using *Pe* = 0.316 for duration of 0.3. This ensures that both factors play a crucial role regarding the migration of NPs towards the interface.

In addition, we have discussed the effect of including or neglecting the particle-particle interaction using the Lennard–Jones potential and we have found that for low concentration of NPs, some differences appear regarding the percentage of NPs within the interface. This is mainly because due to particle–particle interactions NPs close to each other will form clusters and accumulate around each other, so that then NPs first attached to the interface will attract the close ones to accumulate around them so that we find clusters of NPs attached to the interface. In addition, for the same case we find that some clusters will be attached between two close interfaces which match with some experimental observations [14]. On the other hand, we have found that for the case of high concentration of NPs there are several observations depending on the value of the L.J. potential well. For ϵ=1 the particle–particle interaction will be strong enough and will compete the effect of shear and thus the NPs will not be driven to the interface. However, for ϵ=0.1 the NPs will be driven to the interface, where they form a layer at the interfacial region, with some of them are connected with other NPs on adjacent interfaces. From the experimental point of view, it is well known that we can modify the value of the particle–particle potential interactions in order to match some industrial needs and thus the NPs can be driven under the effect of shear to the interface even though when their concentration in the fluids is high. So for this case, we have shown that shearing is a key factor in enhancing the migration of the NPs towards the interface.

On the other hand, we have found that NPs introduced randomly into the mixture of the two fluids before phase separation takes place will migrate faster and in a higher percentage to the interface in the absence of shear.

It is clear that the time of NPs introduction is important. Since the percentage of NPs adsorbing at the interface is less when the phase separation already occurred, seems to indicate that the potential of adsorption is higher in the case. Whereas, when we start from the homogenous state, the potential barrier is less, thus, more particles can be adsorbed at the interface.

These results help us to understand how NPs behave under standard industrial processing conditions. We obtain more information regarding new methods for synthesizing nanomaterial films; in addition, these results also help to understand the behavior of small NPs within the cells of organisms where they are greatly affected by the flow rate of the surrounding fluids.

## Figures and Tables

**Figure 1 entropy-23-01143-f001:**
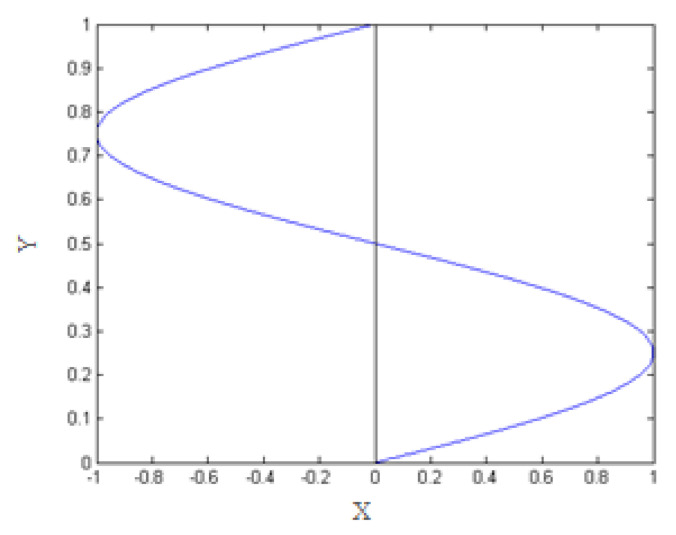
Periodic shear force along the y-axis.

**Figure 2 entropy-23-01143-f002:**
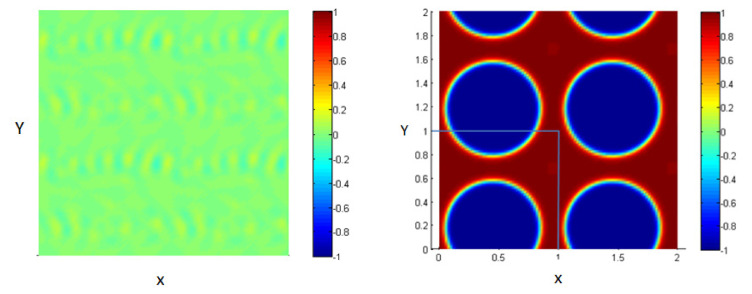
Mixture of two immiscible liquids, from the homogenous state (**left**) to the equilibrium state after the phase separation (**right**).

**Figure 3 entropy-23-01143-f003:**
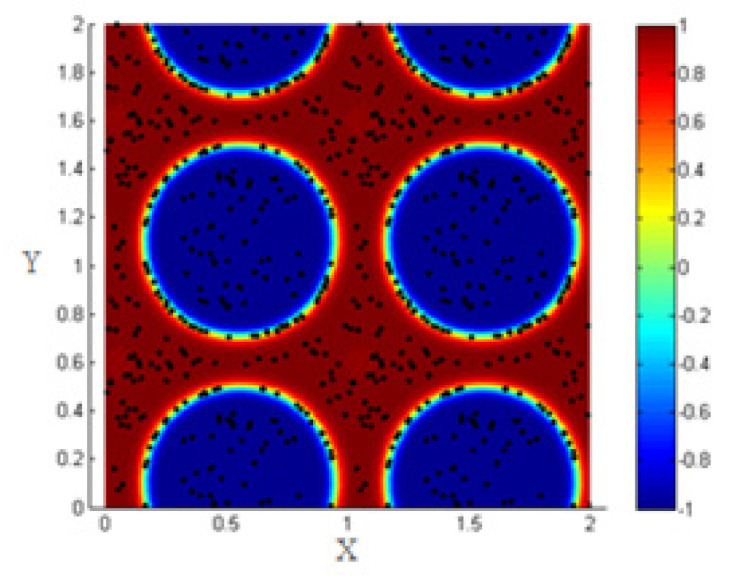
Random distribution of 200 NPs within the two fluids after phase separation.

**Figure 4 entropy-23-01143-f004:**
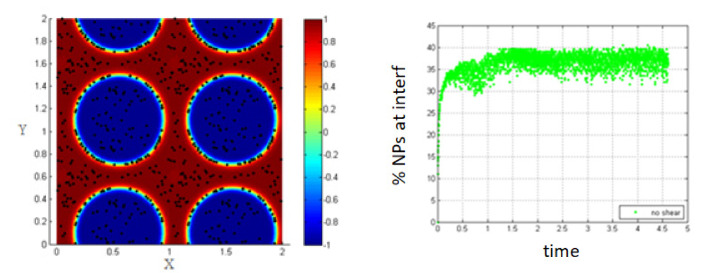
NPs near the interface migrate to the interfacial region (**left**); percentage of NPs belonging to the interfacial region in the absence of shear (**right**).

**Figure 5 entropy-23-01143-f005:**
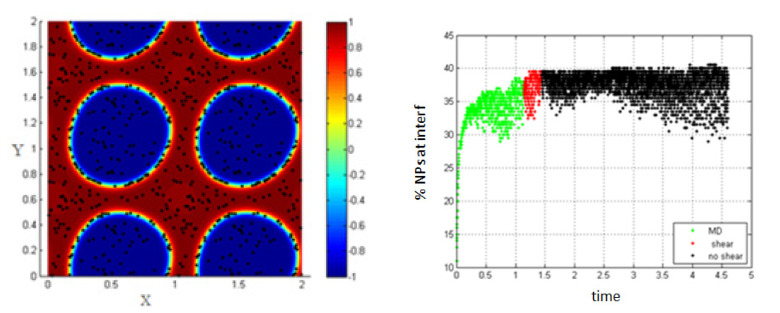
Shape deformation of the two fluids under the effect of shear (**left**); percentage of NPs belonging to the interfacial region in the regions, before applying shear (green), during applying shear (red), and after stopping shearing (black), *Pe* = 0.008 (**right**).

**Figure 6 entropy-23-01143-f006:**
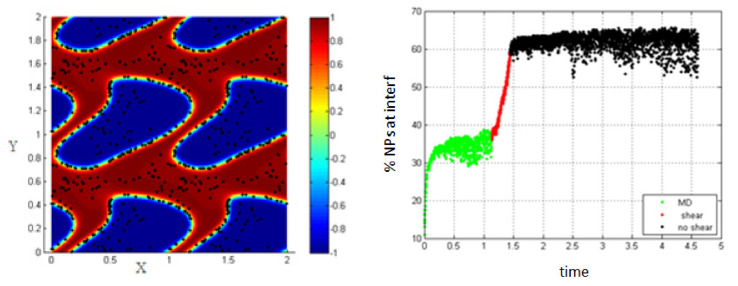
Percentage of NPs belonging to the interfacial region in the regions, before applying shear (green), during applying shear (red) and after stopping shearing (black), *Pe* = 0.154.

**Figure 7 entropy-23-01143-f007:**
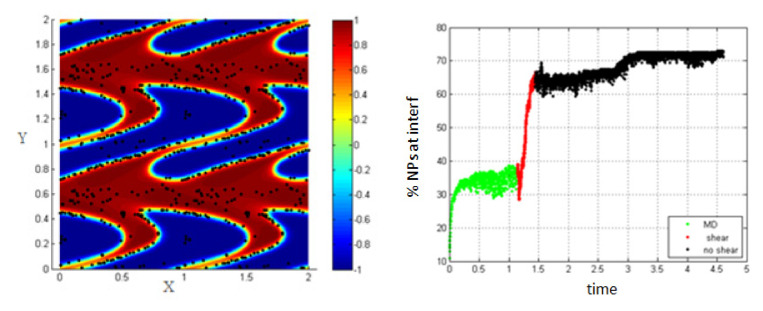
Percentage of NPs belonging to the interfacial region in the regions, before applying shear (green), during applying shear (red) and after stopping shearing (black), *Pe* = 0.316.

**Figure 8 entropy-23-01143-f008:**
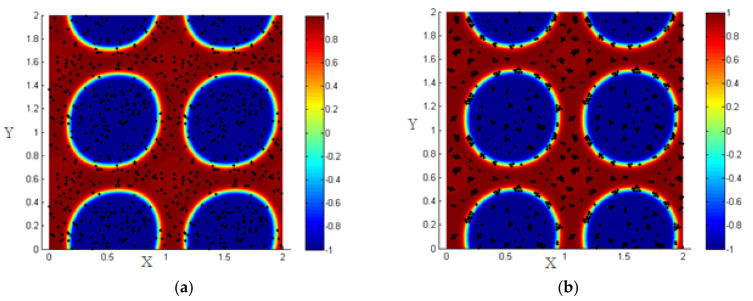
(**a**) Random distribution of the NPs within the two fluids; (**b**) formation of clusters under the effect of particle–particle interactions.

**Figure 9 entropy-23-01143-f009:**
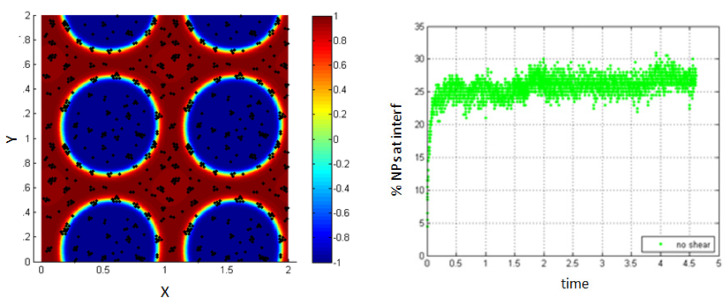
NPs near the interface migrate to the interfacial region (**left**); percentage of NPs belonging to the interfacial region in the regions, before applying shear (**right**).

**Figure 10 entropy-23-01143-f010:**
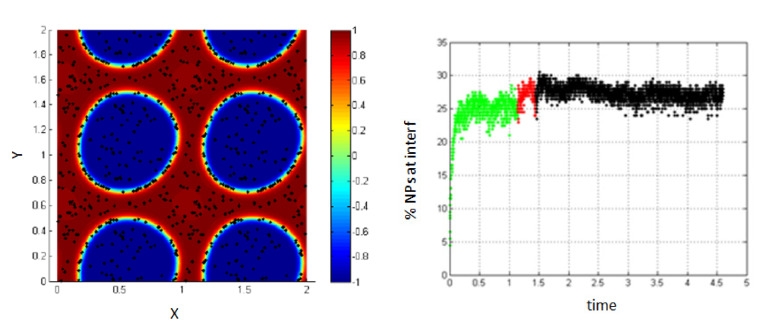
NPs near the interface migrate to the interfacial region (**left**); percentage of NPs belonging to the interfacial region in the regions, before applying shear (green), during applying shear (red) and after stopping shearing (black), Pe = 0.008 (**right**).

**Figure 11 entropy-23-01143-f011:**
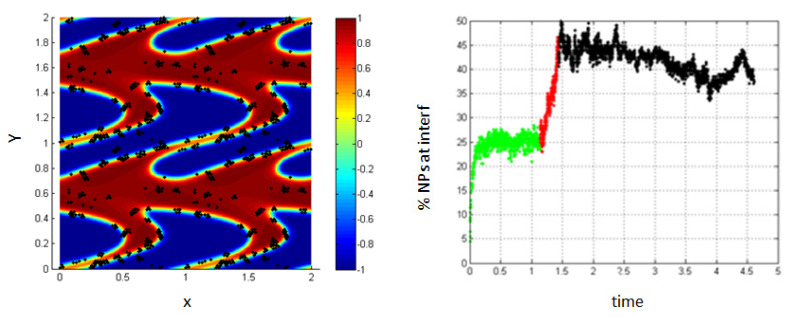
Percentage of NPs belonging to the interfacial region in the regions, before applying shear (green), during applying shear (red) and after stopping shearing (black), Pe = 0.316.

**Figure 12 entropy-23-01143-f012:**
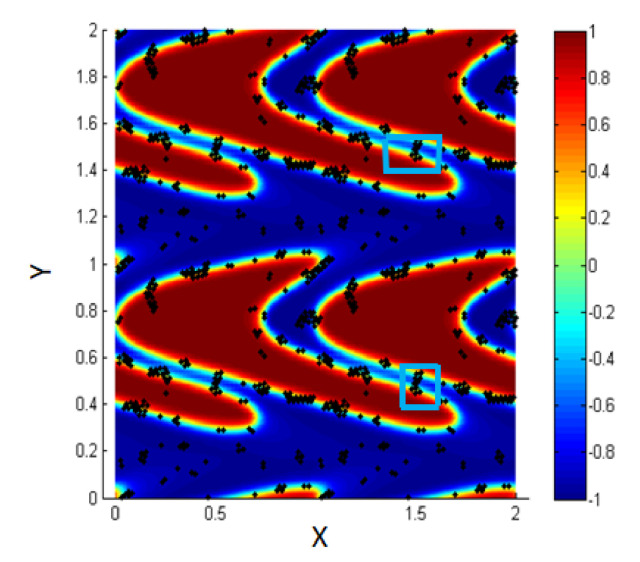
Some clusters are attached between two adjacent interfaces, *Pe* = 0.316.

**Figure 13 entropy-23-01143-f013:**
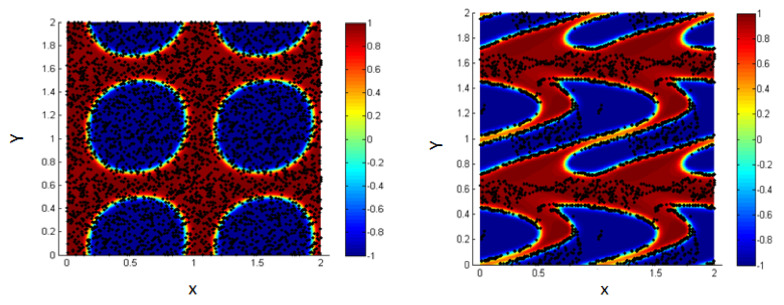
Random distribution of the NPs within the two fluids (**left**); more NPs migrate to the interface under the effect of shear, *Pe* = 0.316, T-shear = 0.3 (**right**).

**Figure 14 entropy-23-01143-f014:**
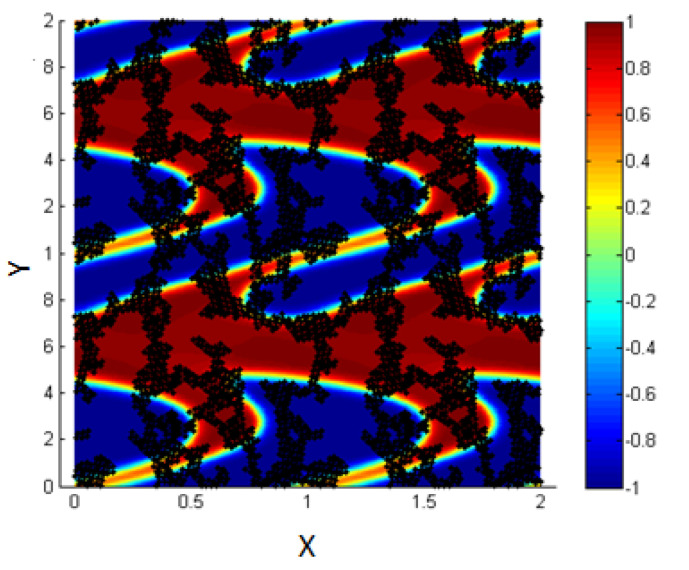
Particle–particle interactions are stronger than the effect of shear.

**Figure 15 entropy-23-01143-f015:**
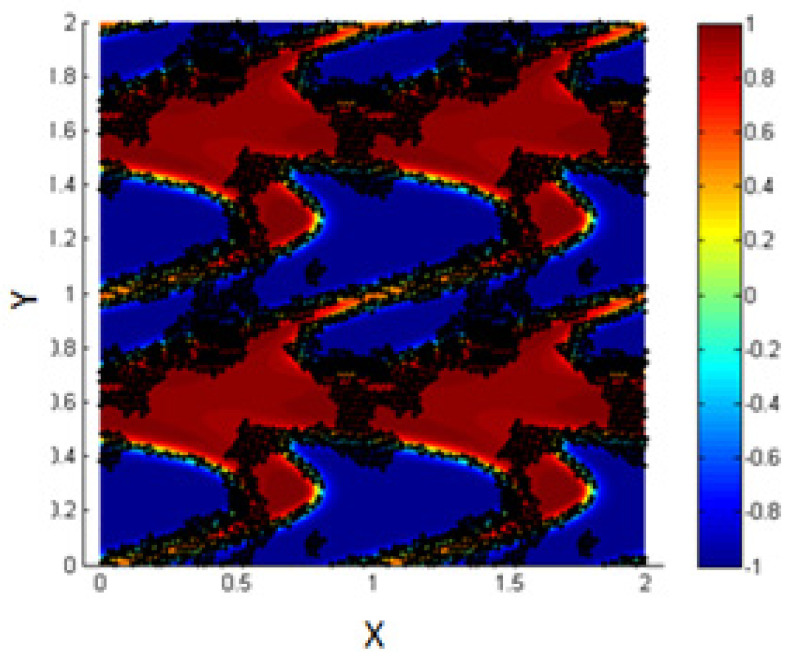
Migration of the NPs clusters to the interface under the effect of shear.

**Figure 16 entropy-23-01143-f016:**
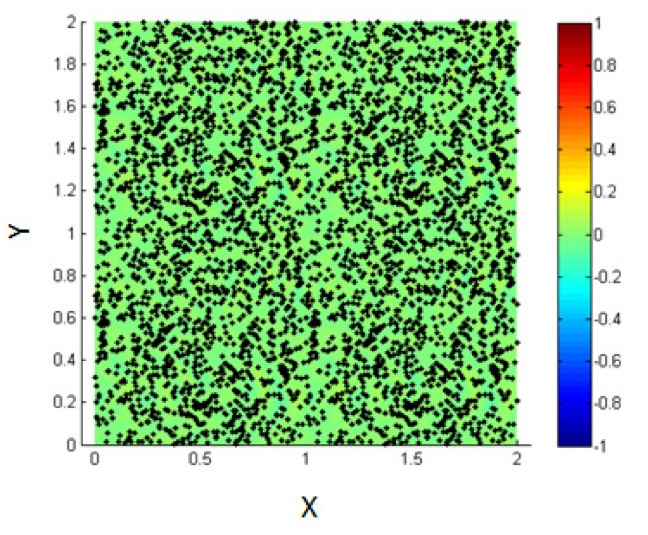
Initial state of NPs randomly distributed within the mixed fluids.

**Figure 17 entropy-23-01143-f017:**
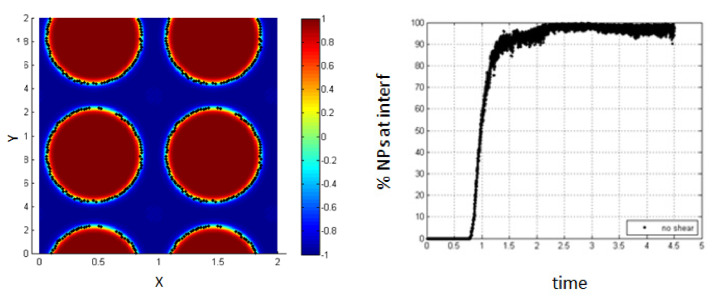
Percentage of NPs belonging to the interfacial region, in the absence of shear.

**Figure 18 entropy-23-01143-f018:**
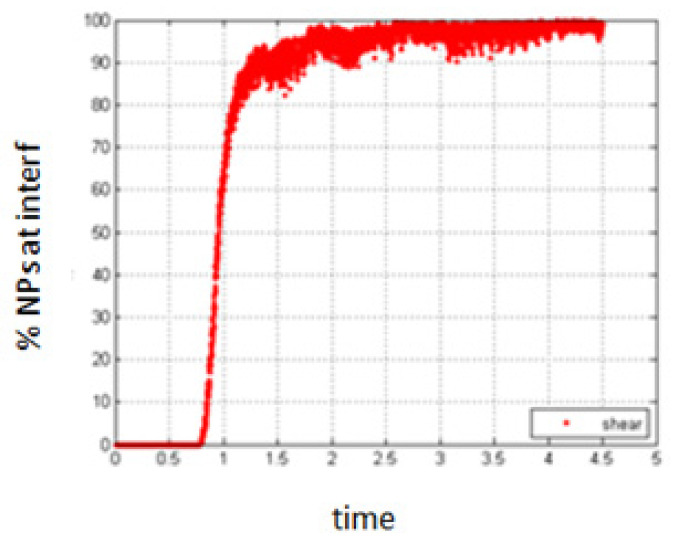
Percentage of NPs belonging to the interfacial region, in the presence of small shear.

**Figure 19 entropy-23-01143-f019:**
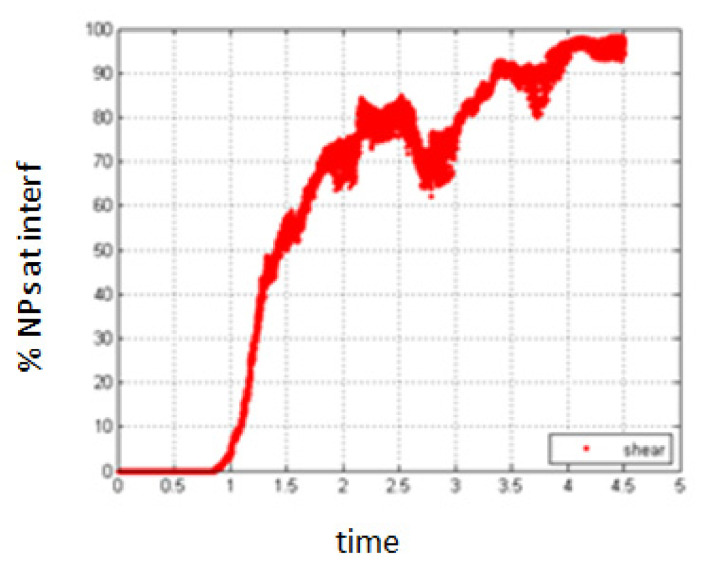
Percentage of NPs belonging to the interfacial region in the presence of shear, *Pe* = 0.154.

**Figure 20 entropy-23-01143-f020:**
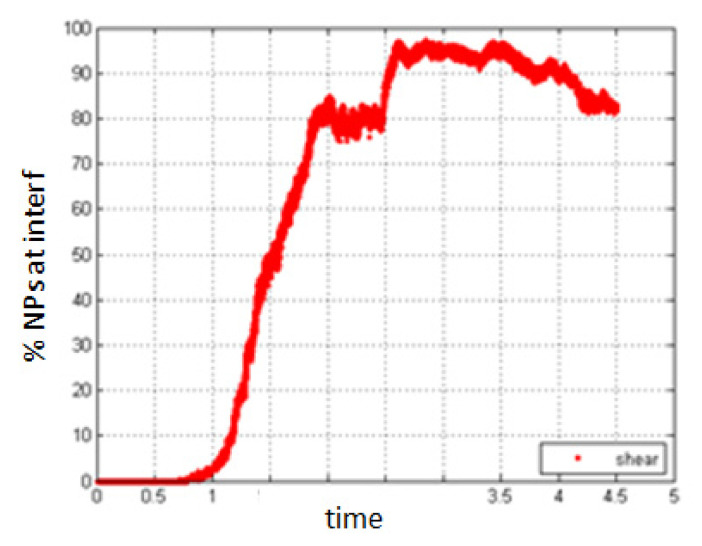
Percentage of NPs belonging to the interfacial region in the presence of shear, *Pe* = 0.316.

**Table 1 entropy-23-01143-t001:** Variation of concentrations through the medium of the two fluids.

Concentration “C”	Liquid 1	Liquid 2
−1	100%	0%
1	0%	100%
0	50%	50%

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
