# Peer review of "Effect of Shear Flow on Nanoparticles Migration near Liquid Interfaces"

_entropy, 2021, doi:10.3390/e23091143_

Round 1
Reviewer 1 Report
The work is nicely written and brings new insights on the liquid-liquid interface for NP, I recommend the manuscript for publication when the authors comment on the questions properly.
- Modeling 41 the dynamics of NPs at liquid –liquid interfaces, has a crucial role in developing static and dynamic flow models that 42 help in drug delivery and understanding the biological and physical phenomena inside the cells of the body. I recommend the authors to mention the dynamics of NP in non-aqueous media such as ionic liquids as it is reported in many studies such as, The application of ionic liquids in nanotechnology ( https://doi.org/10.1016/B978-0-323-51255-8.00012-4), Metal Nanoparticles in Ionic Liquids.( https://doi.org/10.1007/s41061-017-0148-1), Static and dynamical properties of colloidal silver nanoparticles in [EMim][PF6] ionic liquid (DOI: 10.1515/9783110583632-006), Effect of Nitrogen Doping on Glass Transition and Electrical Conductivity of [EMIM][PF6] Ionic Liquid Encapsulated in a Zigzag Carbon Nanotube (DOI: 10.1021/acs.jpcc.7b00911)
- The concentration is taken to vary from (-1) in the first fluid (blue) to (1) in the second one (red). Which ratio has been used for two liquids? 1:1 ratio or different? Does it have any influence on the phase separation?
- After the two fluids have been separated and the interface has been formed between them, we are going to introduce N 301 nanoparticles randomly into the system. We will consider two cases regarding the concentration of NPs . As two liquids have different affinity to NP, when you add the NP before phase separation to the medium why the migration of NPs can be influence?.
- Figure 4. NPs near the interface migrate to the interfacial region (left); Percentage of NPs belonging 329 to the interfacial region in the absence of shear (right). In the graph ( right) time is mentioned, what is the scale of time, ps, ns?
- How is the distribution of concentration along the medium from bulk to the liquid – liquid interface? It can be calculated as density profile. Also in Figure 5. The time scale is missing.
- In Figure 7, It is clear that increasing the shear rate and thus increasing the Péclet number, will enhance the migration 380 of the NPs towards the interface, to reach about 62% at the end of the shear and also this percentage increases a little bit 381 to reach 70 % after stopping shearing. Why?
- In reality they agglomerate around the ones that are in the interfacial thickness so that the cluster will be attached to the 420 interface by some of the NPs forming it. It will be good idea to show the distribution of NP in the bulk and at the interface to prove your hypothesis.
- These results match some experi-460 mental results through which particles agglomerate at the interface and sometimes are attached between two close 461 interfaces as shown in Figure 12. Any reference?
- MD simulations methodology needed to be explained in more detail.
Reviewer 2 Report
The article deals with a very interesting phenomenon, the effect of shear flow on nanoparticle motion near two immiscible liquids. The text is well structured and easy to read. Even for the non-specialist the article is well understandable. This was made possible by the clear structure and the recurring and constant sequence of the text structure (for this few comments below). The figures are almost self-explanatory and the simulation results are well presented in the summary.
Comments:
104: The order of the formula description should follow the writing direction. For this, the definition of rc would still be for ε.
290-291: The definition of the axes should be given to the image description.
314: A paragraph before the heading is missing here.
328: The font size of the axis caption in the right picture should be larger. This also applies to the following figures.
345: The abbreviation MD in figure 5 should be more clearly explained or visible in the text. Likewise, the reference to figure 5 is missing in the text, even if it is not absolutely necessary. The same applies to figures 9 and 10.
416: For the purpose of uniformity, a simulation image with the particles would also be nice for figures 9 and 10.
475: To increase clarity, one can mark the area in figure 12 where the two interfaces come so close that the particles accumulate between them.
520: sigma should be added in analogy to line 512.
